# Effect of Ball-Milled Feedstock Powder on Microstructure and Mechanical Properties of Cu-Ni-Al-Al$_2$O$_3$ Composite Coatings by Cold Spraying

**Hongjin Liu** [1,2,3], **Mingkun Fu** [1,2,3], **Shaozhi Pang** [1,2,3], **Huaiqing Zhu** [1,2,3], **Chen Zhang** [1,2,3], **Lijun Ming** [1,2,3], **Xinyu Liu** [1,2,3], **Minghui Ding** [1,2,3,*] and **Yudong Fu** [1,2,3]

1   College of Material Science and Chemical Engineering, Harbin Engineering University, Harbin 150001, China;
    b9838@126.com (H.L.); fmk2002@163.com (M.F.); psz18845643023@163.com (S.P.);
    17661047827@163.com (H.Z.); zhang.chen@hrbeu.edu.cn (C.Z.); minglijunabc@163.com (L.M.);
    13114593371@163.com (X.L.); fuyudong@hrbeu.edu.cn (Y.F.)
2   Key Laboratory of Super Light Material and Surface Technology Ministry of Education, Harbin Engineering
    University, Harbin 150001, China
3   Institute of Surface/Interface Science and Technology, Harbin Engineering University, Harbin 150001, China
*   Correspondence: mhding@hrbeu.edu.cn; Tel./Fax: +86-451-8251-8219

**Abstract:** Cu, Ni and Al powders mixed in a certain stoichiometric proportion were ground via ball milling and deposited as coatings using low pressure cold spraying (LPCS) technology. The effect of particle morphology on the powder structure as well as the microstructure, composition and mechanical properties of the coatings was studied. The results revealed a core–shell structure of ball-milled powders. Compared with a mechanically mixed (MM) coating, coatings after ball milling at a rotation speed of 200 rpm exhibited the most uniform composition distribution and a lower degree of porosity (by 0.29%). Moreover, ball milling at 200 rpm was conducive to a significant increase in the deposition efficiency of the sprayed powder (by 10.89%), thereby improving the microhardness distribution uniformity. The ball milling treatment improved the adhesion of the coatings, and the adhesion of the composite coating increased to 40.29 MPa with the increase in ball milling speed. The dry sliding wear tests indicated that ball milling treatment of sprayed powder significantly improved the wear properties of the coatings. The coating after ball milling at a speed of 250 rpm showed the lowest friction coefficient and wear rate, with values of 0.41 and 2.47 × 10$^{-12}$ m$^3$/m, respectively. The wear mechanism of coatings changed from abrasive wear to adhesive wear with the increase in ball milling speed.

**Keywords:** ball milling; low-pressure cold spraying; Cu-based composite coatings; microstructure; deposition efficiency; mechanical properties

## 1. Introduction

Low-pressure cold spraying (LPCS) is a coating preparation technology based on supersonic fluid dynamics and high-speed impact dynamics [1–4]. Thanks to easy implementation and high efficiency, the method has broad application prospects in the fields of device repair, remanufacturing and additive manufacturing [5–7]. During spraying, the powder particles are accelerated to a supersonic speed under the action of a certain gas pressure, exerting influence on the surface of the substrate to produce severe plastic deformation and to form a coating [8–10]. Compared with traditional thermal spraying methods, the LPCS has the advantages of lower carrier gas temperature and less pronounced thermal impact on the substrate, thereby preventing the oxidation processes and ensuring low porosity and strong bonding between the particles in the coating [11–13]. Therefore, LPCS is suitable for depositing phase change sensitive materials and oxidizable materials [14,15].

The structure and properties of LPCS composite coatings can be adjusted and controlled via the following stages: the preparation of composite powders before spraying, the

mixture of powders during spraying and the post-treatment of coatings (e.g., rolling, heat treatment, etc.). In particular, the morphology and structure of spraying powders are the important factors affecting the structure and performance of LPCS-produced coatings [16–18]. A thoroughly prepared powder can improve the deposition efficiency, structure and performance of the coating, which is essential for further repairing and remanufacturing using LPCS [19–21]. The common preparation routes of composite powders include mechanical mixing, spray drying, ball milling and sintering [22–27]. For example, Xiao [28] obtained a core–shell-structured WC-Co powder via ball milling and deposited it as a coating that possessed a uniform and dense structure with a porosity of 0.7% only. Li [29] produced a tin-reinforced Al5356 coating through ball milling. In all cases, ball-milled (BM) powders endowed the coatings with a denser and more uniform structure as well as a better wear resistance.

Cu/Ni/Al composite coating is a relatively complex coating system, there are many combinations of different materials (such as Cu/Al, Ni/Al, Cu/Ni), which are helpful for studying the structure and performance of coatings in different systems. The dispersion degree and morphology of spray powder have a significant impact on the structure and performance of the coatings. Therefore, this work aims to study Cu-Ni-Al-$Al_2O_3$ coatings fabricated via LPCS so as to establish the effect of BM powder morphology on their structure, morphology and mechanical properties.

## 2. Experimental Procedures

Commercial Cu (20–25 μm), Ni (20–25 μm) and Al (25–30 μm) powders were used as raw materials. Powders (56 wt.% Cu), (24 wt.% Ni) and (20 wt.% Al) were mechanically mixed and ball-milled for 4 h under an Ar atmosphere with a planetary ball mill (UBE-F2L, China) using ZrO grinding balls (10 mm, 5 mm and 3 mm). The rotation speeds were set to 150 rpm, 200 rpm and 250 rpm, and the ball-to-powder mass ratio was 10:1. To avoid excessive temperature rise during ball milling, the procedure was suspended for 10 min every half hour.

A LPCS system (TCY-LP-III, Beijing Tianchengyu New Material Technology Co., Ltd., Beijing, China) was employed for coating preparation. Before spraying, powders were mechanically mixed with $Al_2O_3$ powder (45–50 μm) at a mass ratio of 7:3 to improve the deposition efficiency and coating density [30,31]. Compressed air was used as accelerating gas at a pressure of 0.8–1.0 MPa and a temperature of about 550 °C. A standoff distance from the nozzle exit to the substrate surface was 15 mm. AZ91D magnesium alloy served as the substrates. Before powder spraying, the substrates were exposed to ultrasonic cleaning for 10 min with deionized water, acetone and ethanol. After that, blow drying and carborundum abrasion were performed to blast the cleaned substrates and roughen their surfaces so as to remove the oxide layer and allow the easier powder deposition [14]. In this paper, the Cu-based coatings were polished before testing.

A scanning electron microscope (SEM) (JSM-6480A, JEOL Ltd., Tokyo, Japan) was used to observe the surfaces and cross-sections of the composite coatings and the morphology of the frictions and wear areas. The energy dispersive spectrometer (EDS) coupled with the SEM instrument enabled the analysis of the element contents in the coatings. The images were acquired at the operating voltage of 20 kV and processed in image J software to assess the particle size of the powder and the porosity of coatings (porosity was measured using the Threshold function of image J software).

X-ray diffraction (XRD) (X'Pert Pro, PANalytical B.V., Almelo, the Netherlands) was applied to analyze the phase compositions of sprayed powders and Cu-based composite coatings. The measurements were carried out using a Cu target at the voltage of 40 kV and the current of 40 mA. The XRD profiles were calibrated by means of HighScore software and standard PDF cards.

A total of 20 g of spray powder was weighed and sprayed onto the AZ91D magnesium alloy substrate, and the powder deposition efficiency was calculated using the following formula:

$$DE = \frac{M_1 - M_2}{M_3} \times 100\% \tag{1}$$

where $DE$ is the deposition efficiency of the powder; $M_1$ is the weight of the sample after spraying (g); $M_2$ is the weight of AZ91D magnesium alloy substrate (g); and $M_3$ is the weight (g) of the sprayed powder weighed.

The adhesion of coatings was measured on an electronic universal testing machine (WDW-100, Jinan Fangyuan Testing Instrument Co., Ltd., Jinan, China). Prior to the experiment, the mixed powder was sprayed onto the cylindrical AZ91D magnesium alloy base with a diameter of 20 mm. The obtained coating was then polished and bonded to another cylindrical magnesium alloy base. The pull-out test of the bonded samples was afterward carried out at a tensile speed of 0.2 mm/min, and the load corresponding to the coating pull-off was recorded. Each group of tests was repeated three times and the average value was taken. The bonding strength of the coatings was determined as follows:

$$\sigma = \frac{F}{A} \tag{2}$$

where $\sigma$ is the bonding strength of the coating (MPa); $F$ is the destructive force at which the coating is broken (N); and $A$ is the area of the sample column (mm$^2$).

The microhardness of the Cu-based composite coatings was evaluated on a microhardness tester (HVS-1000B, Laizhou Huayin Testing Instrument Co., Ltd., Yantai, China) under a load of 100 g applied for 15 s. The average value was found after five hardness measurements.

A high-temperature friction and wear tester (HT-1000, Lanzhou Zhongke Kaihua Technology Development Co., Ltd., Lanzhou, China) was used to determine the wear resistance of coatings. The friction pair in the test was a GCr15 grinding ball with a diameter of 6 mm. During the experiment, the friction pair moved in a circular direction with a radius of 3 mm at a rotation speed of 280 r/min. The load was 5 N, and the test time was 10 min. After the completion of the friction and wear tests, the wear marks left on the sample were observed by SEM and their shape was assessed as well. At the same time, the volume wear rate of the samples was calculated from the formulae below [32]:

$$\Delta V = L0(r^2 \arcsin\frac{d}{2r} - \frac{d}{2}\sqrt{r^2 - (\frac{d}{2})^2}) \tag{3}$$

$$W_N = \frac{\Delta V}{L} \tag{4}$$

where $W_N$ is the volume wear rate of composite coatings (m$^3$/M); $\Delta V$ is the volume loss of composite coatings (m$^3$); $L$ is the friction distance in the test (m); $L0$ is the circumference of the wear mark (m); $r$ is the radius of the friction pair (m); and $d$ is the wear mark width (m).

## 3. Results and Discussion

### 3.1. Characterization of Spray Powders

Figure 1 displays the microscopic topography of the sprayed powder after ball milling. According to Figure 1a, the powder at the rotation speed of 150 rpm still retained its original appearance and only a few particles were extruded and deformed therein. However, once the speed further increased to 200 and 250 rpm, the particles agglomerated and became irregular (Figure 1b,c). Figure 1d,e depicts the cross-sectional microstructures of sprayed powders after milling. It was found that powders possessed a core–shell structure, in which the Cu and Ni cores were covered by the Al shell. This could be attributed to the fact that during the ball milling, the softer Al powder continuously accumulated on the surface of the Cu and Ni powders to form a core–shell-structured powder. With the increase in rotation speed, the particle size in the powders decreased first and then increased. At

the speed of 200 rpm, the particle size of the powder was about 13 μm. As soon as the speed increased to 250 rpm, the particle size approached 30 μm. At the same time, the particle shape became more irregular, which indicated that the particles underwent strong deformation and aggregation during milling.

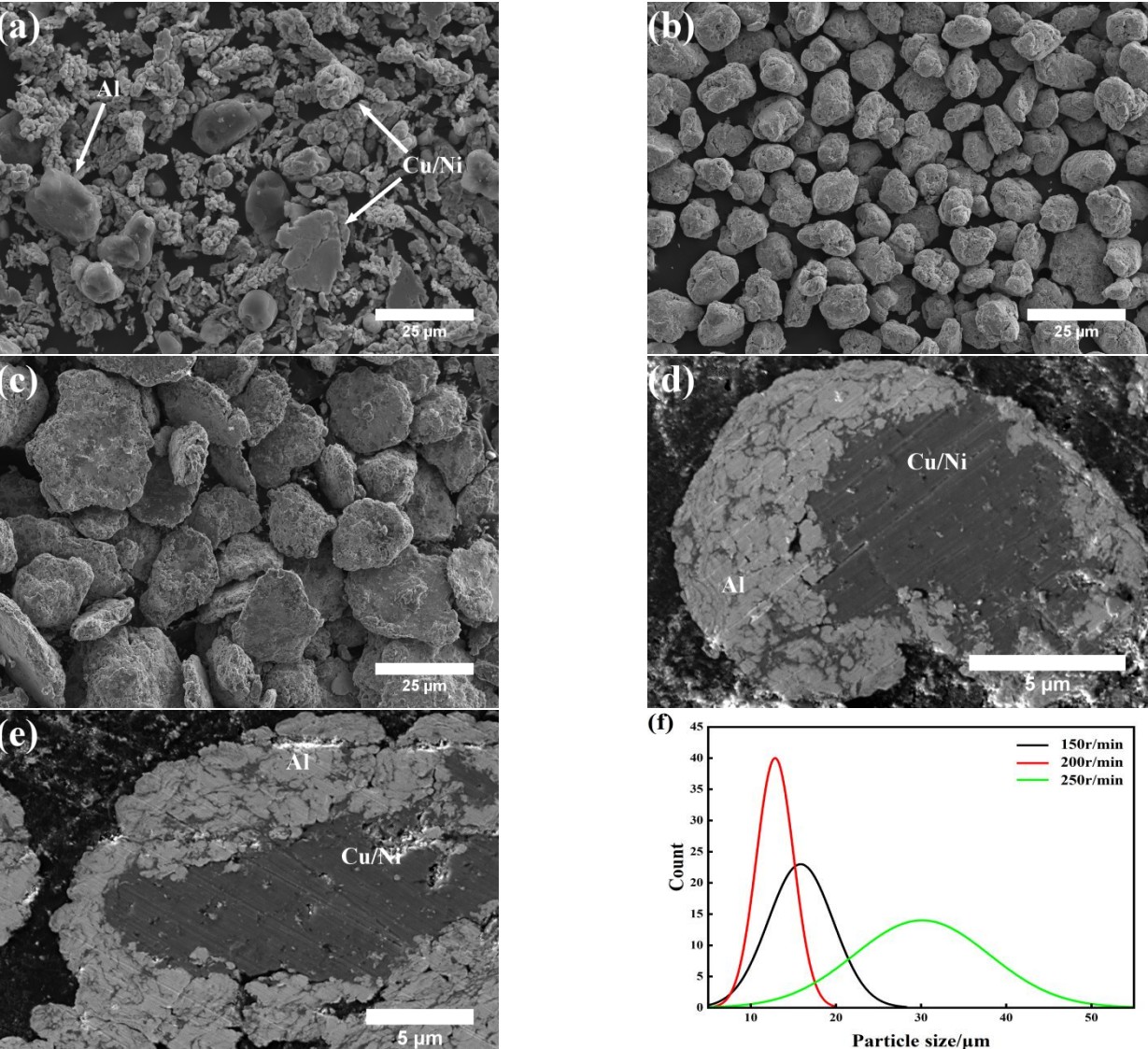

**Figure 1.** SEM images of powders at the rotation speed of (**a**) 150 rpm, (**b**) 200 rpm and (**c**) 250 rpm. (**d**) Cross-sectional microstructure of powder milled at 200 rpm. (**e**) Cross-sectional microstructure of powder milled at 250 rpm. (**f**) Particle size distributions in spray powders.

Figure 2 displays the XRD results of powders after ball milling at different rotation speeds. In all cases, the XRD profiles were quite similar to each other, meaning that the ball milling basically did not alter the phase structure of powders. A comparison of these XRD spectrograms with the XRD database (JCPDS cards nos. 00-004-0836, 00-004-0850, and 96-900-8461) revealed a stable presence of Cu, Ni and Al phases. Therefore, the ball milling could have only impacted the microstructure of powders conforming to the SEM images in Figure 1.

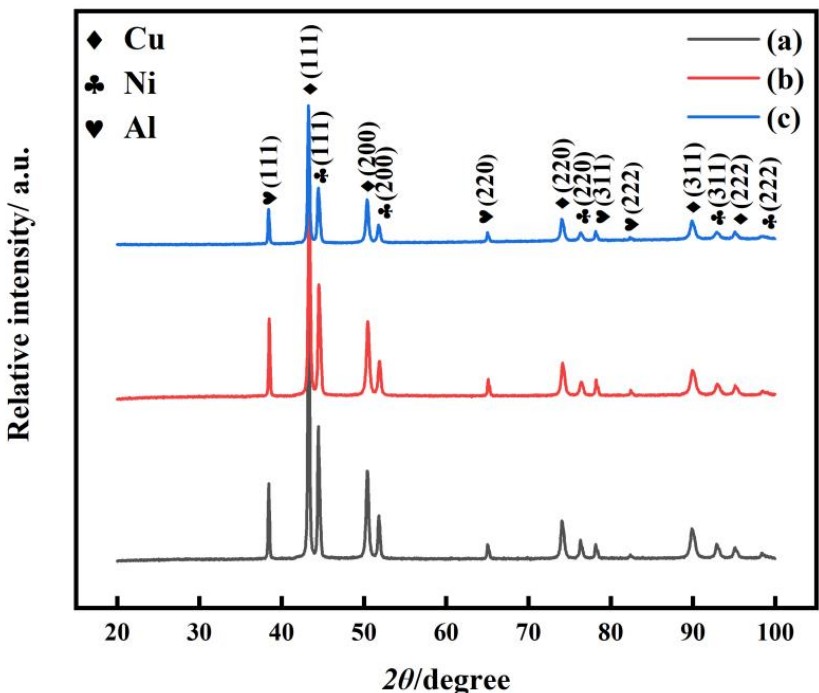

**Figure 2.** XRD pattern of spray powders at different rotation speeds: (**a**) 150 rpm, (**b**) 200 rpm and (**c**) 250 rpm.

### 3.2. Microstructures of Coatings

Figure 3 depicts the SEM images of coatings. In the mechanically mixed (MM) coating, the phases were homogeneously distributed (Figure 3a). In turn, the constituent phases in the ball-milled (BM) coatings were distributed in a more uniform manner, changing from isolated to staggered configurations (Figure 3c,d) because of the core–shell structure of powders. Figure 4 depicts the cross-sectional SEM images of coatings. In each case, the bonding interface between the coating and the substrate in the form of an irregular curve could be clearly observed. It was attributed to the severe plastic deformation of the powder particles after they collided with the substrate at the high speed and then combined together with the substrate.

Combining the SEM images of the surface and the cross-section images of coatings (Figures 3 and 4), it was implied that powders were firmly bonded to the substrates. No obvious pores and cracks were detected at the bonding interfaces and within the coatings. The overall porosity of the coatings was less than 1%. According to Table 1, the porosity at the rotation speed of 200 rpm was 0.29% (Table 1). However, once the speed rose to 250 rpm, the porosity increased to 0.76% (Table 1), which could be attributed to the fact that the particle size increased and the morphology became flat (Figure 1c,f), making the powder unsuitable for spraying and thereby reducing the density of the coating. At the same time, scarce $Al_2O_3$ particles were embedded in the coating, which played the role of compaction and secondary shot peening during the LPCS, thus reducing the porosity and increasing the compactness of the coating. However, while they possessed high hardness, the $Al_2O_3$ particles lacked any deformation ability and could not match the sprayed powder, causing the pore concentrations around them.

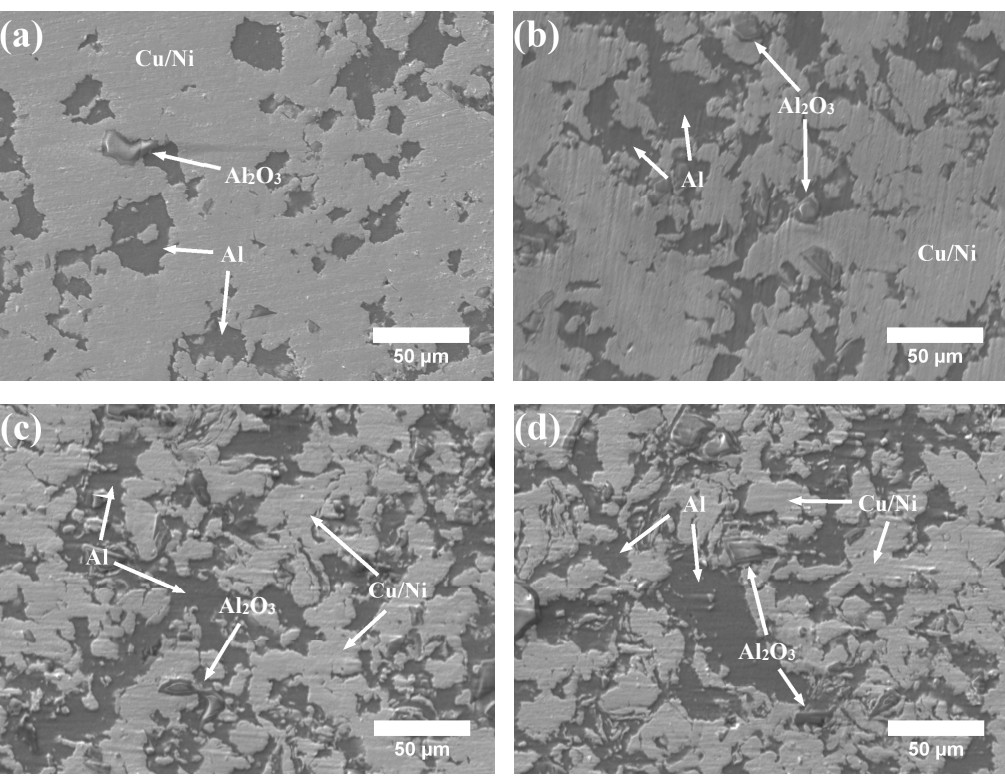

**Figure 3.** SEM images of coatings: (**a**) MM powder, (**b**) BM powder milled at 150 rpm, (**c**) BM powder milled at 200 rpm and (**d**) BM powder milled at 250 rpm.

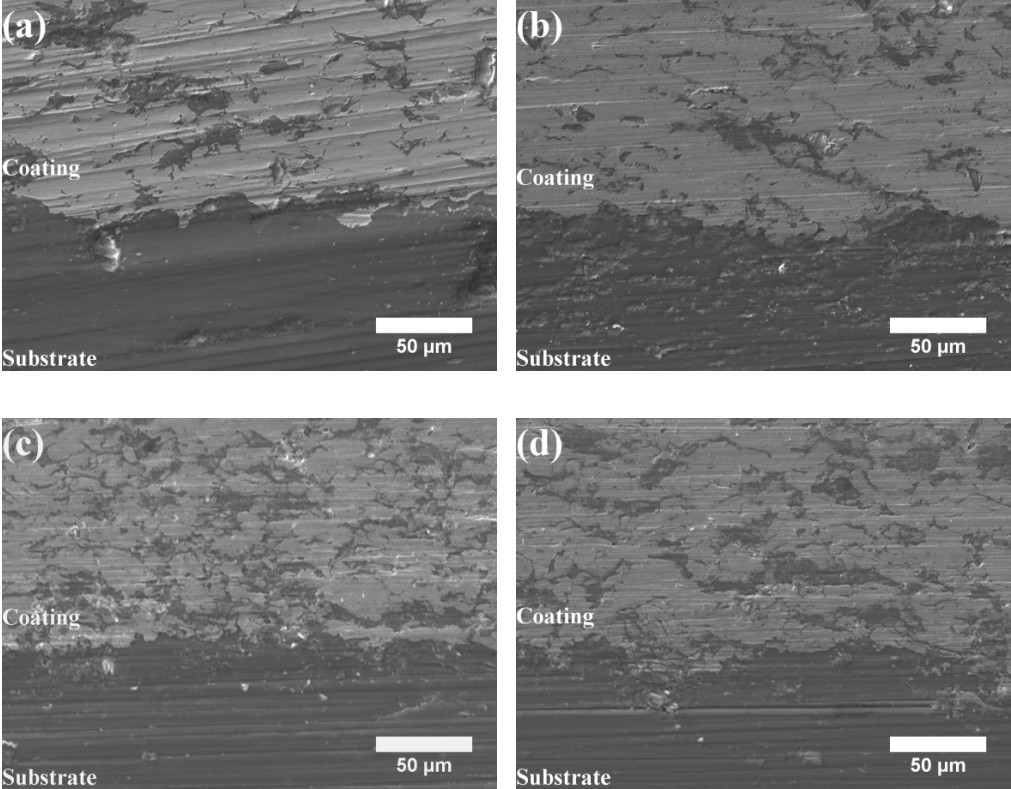

**Figure 4.** Cross-sectional SEM images of the coatings: (**a**) MM powder, (**b**) BM powder milled at 150 rpm, (**c**) BM powder milled at 200 rpm and (**d**) BM powder milled at 250 rpm.

**Table 1.** Material content and porosity of coatings.

| Deposit | Cu (wt.%) | Ni (wt.%) | Al (wt.%) | Al$_2$O$_3$ (wt.%) | Porosity (%) |
|---|---|---|---|---|---|
| MM | 84.2 | 9.34 | 4.82 | 1.65 | 0.58 |
| BM (150 rpm) | 70.54 | 18.46 | 8.72 | 2.28 | 0.41 |
| BM (200 rpm) | 63.17 | 19.70 | 14.39 | 2.74 | 0.29 |
| BM (250 rpm) | 59.45 | 12.01 | 26.30 | 2.24 | 0.76 |

Table 1 summarizes the EDS results on the coatings. The mass fraction of Al$_2$O$_3$ was calculated by using the mass fraction of O element. The mass fraction of Al$_2$O$_3$ should be slightly lower than the calculated value because a small amount of Al was oxidized during spraying. It was obvious that the BM powder increased the contents of Ni and Al in the coatings. Compared with Cu, the higher hardness of Ni made it difficult to deposit, while the smaller density of Al led to its lower kinetic energy during spraying, which was also not conducive to spraying. After the ball milling, on the one hand, the amounts of Ni and Al in the powder with a core–shell-structure dramatically increased during the co-deposition process; on the other hand, the Al shell strongly bonded to the substrate, which could make the coatings more compact, according to the porosity analysis.

Figure 5 depicts the XRD patterns of the coatings prepared from MM and BM powders at different rotation speeds. According to the data, the phase structures of the coatings were consistent with those of the sprayed powders (Figure 2), revealing neither oxidation nor phase transformation during the LPCS, as expected. Meanwhile, no diffraction peaks of Al$_2$O$_3$ appeared, indicating that a small amount of Al$_2$O$_3$ particles remaining in the coatings did not affect their structure.

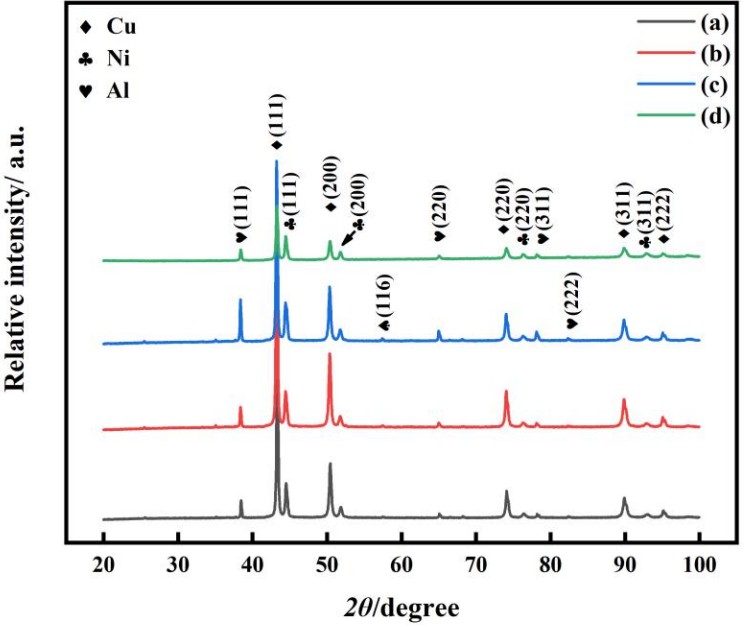

**Figure 5.** XRD patterns of coatings: (**a**) MM powder, (**b**) BM powder milled at 150 rpm, (**c**) BM powder milled at 200 rpm and (**d**) BM powder milled at 250 rpm.

### 3.3. Mechanical Performance of Coatings

As shown in Table 2, compared with the MM powder, ball milling significantly improved the deposition efficiency of the powder. In particular, the deposition rate of the ball-milled powder at the rotation speed of 200 rpm was 10.89% higher than that of the MM powder. The better deposition performance of the powder with a core–shell structure was attributed to the fact that the Cu and Ni cores possessed the sufficient kinetic energies. At the same time, the Al shell could have experienced severe plastic deformation. However,

the deposition efficiency of the sprayed powder decreased to 9.36% only at a rotation speed of 250 rpm, indicating that the powder was not suitable for spraying at this time.

**Table 2.** Deposition efficiency, hardness, adhesion, friction coefficient and volume wear rate of coatings.

| Deposit | Deposition Efficiency (%) | Hardness ($HV_{0.1}$) | Adhesion (MPa) | Friction Coefficient | Wear Rate ($\times 10^{-12}$ m$^3$/m) |
|---|---|---|---|---|---|
| MM Powder | $30.71 \pm 2.13$ | $155.76 \pm 6.71$ | $31.17 \pm 2.93$ | $0.56 \pm 0.051$ | 8.43 |
| 150 rpm | $36.23 \pm 2.49$ | $149.88 \pm 3.21$ | $31.45 \pm 3.03$ | $0.51 \pm 0.045$ | 10.19 |
| 200 rpm | $41.60 \pm 3.02$ | $153.03 \pm 1.34$ | $40.29 \pm 3.95$ | $0.49 \pm 0.035$ | 4.92 |
| 250 rpm | $9.36 \pm 1.31$ | $136.55 \pm 10.00$ | $37.44 \pm 3.18$ | $0.41 \pm 0.024$ | 2.47 |

The hardness of coatings is given in Table 2. It was established that the impact of ball milling on the rigidity of coatings was not obvious, meaning that the work hardening of the powders due to plastic deformation in the deposition process exceeded the effect of ball milling on the powders themselves. Meanwhile, in addition to the work hardening, the powder dispersion uniformity also exerted strong influence on the hardness of the coating. Figure 6 displays the hardness through the specimens as a function of distance from the substrate. It was evident that the hardness of the MM coating and BM (250 rpm) coating fluctuated to a large extent, while slightly changing in BM coatings treated at 150 and 200 rpm. This indicated the improvement in internal uniformity of the two latter coatings. A drastic variation in the hardness of the coating processed at 250 rpm could be ascribed to the excessive aggregation and uneven distribution of Al elements in the outer layer of the powder during ball milling (Figure 3d). At the same time, the high porosity of the coating (Table 1) was another important factor leading to the hardness fluctuations throughout the coating.

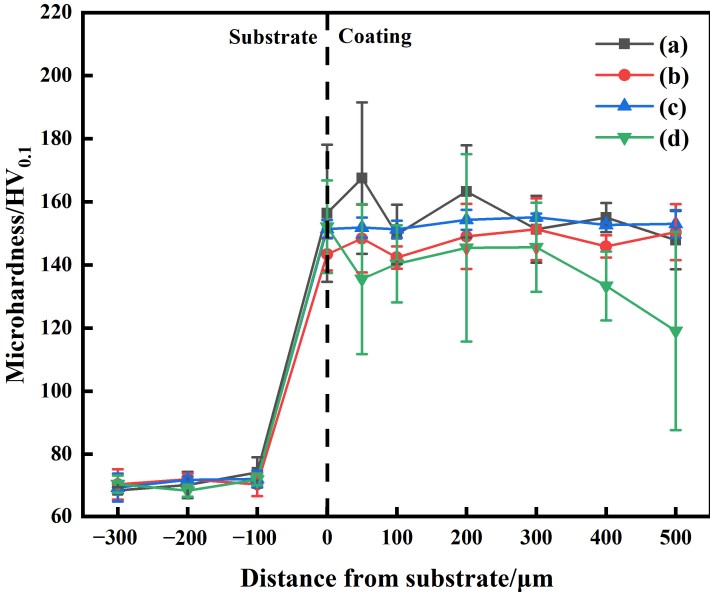

**Figure 6.** Microhardness of coatings: (**a**) MM powder, (**b**) BM powder milled at 150 rpm, (**c**) BM powder milled at 200 rpm and (**d**) BM powder milled at 250 rpm.

The adhesion properties of coatings are shown in Figure 7. It was found that the adhesion of the BM coating treated at 150 rpm was the same as that of the MM coating. With the increase in the rotation speed, the adhesion of coatings increased to a large extent, which was related to the microstructural peculiarities of the relevant powders (Figure 1a–c). In particular, the adhesion of the BM coating was 40.29 MPa at the speed of 200 rpm (Table 2), which was 29.26% higher than that of the MM coating. This was because the core–

shell-structured powders improved the adhesion of coatings due to the stronger mechanical engagement ability between the Al shell and the substrate [33]. In the tensile testing, all the coatings were broken in the middle, indicating that the specimens underwent cohesive failure. In a word, the bonding strength between the coatings and the substrates was higher than the cohesion strength of the coatings themselves.

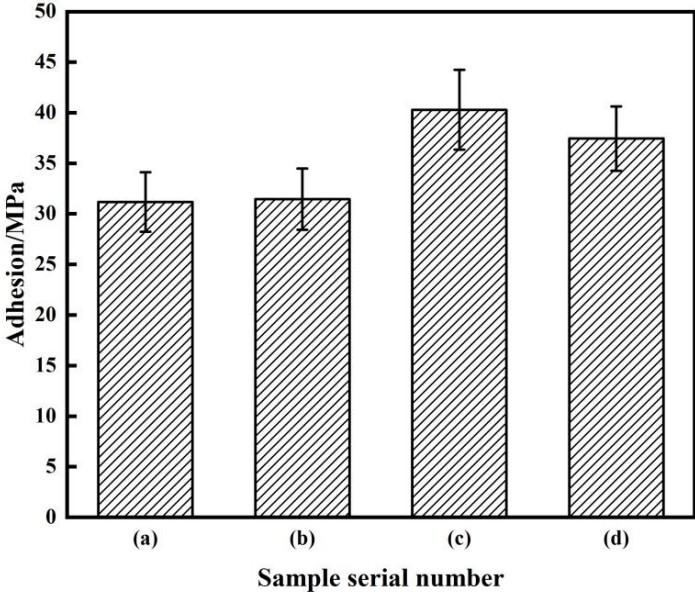

**Figure 7.** Adhesion of coatings: (**a**) MM powder, (**b**) BM powder milled at 150 rpm, (**c**) BM powder milled at 200 rpm and (**d**) BM powder milled at 250 rpm.

Figure 8 depicts the evolution of friction coefficients of composite coatings with the friction time. After a short running-in period (about 1 min), the friction coefficients of coatings reached a relatively stable state. The average friction coefficients estimated from the friction times between 2 and 10 min are listed in Table 2. In particular, the friction coefficients of BM composite coatings were lower than that of the MM coating, which was associated with a more uniform composition distribution in the former coatings, and the higher the rotation speed, the lower the friction coefficient. In addition, the wear rate (WR) of BM coatings decreased with the increase in milling speed. For example, the WR of the coating at 250 rpm was $2.47 \times 10^{-12}$ m$^3$/m, being only about one-third of that of the MM coating, despite the composite coating yielding lower hardness values [34,35]. This means that the ball milling of spraying powders under the optimal conditions were able to significantly improve the wear resistance of coatings and reduce their friction coefficient.

To elucidate the wear mechanisms of the coatings, the corresponding SEM images were further analyzed (Figure 9). The wear surfaces of the BM coating (150 rpm) and MM coating were characterized by deep grooves and scratches (Figure 9a,b). On the other hand, some hard particles peeled off due to wear providing abrasives for further abrasive wear. These particles remained between the friction pair and the coating to cut the coating, thus forming plenty of grooves and furrows parallel to the friction direction on the wear surface. Therefore, both coatings experienced abrasive wear [36]. Once the ball-milling speed reached 200 rpm and 250 rpm, continuous peeling cracks and delamination could be observed, which were attributed to the overall fracture and peeling of the coating material as well as plastic deformation during the friction process, leading to adhesive wear. Therefore, the wear mechanism of coatings changed from abrasive wear to adhesive wear with the increase in rotation speed.

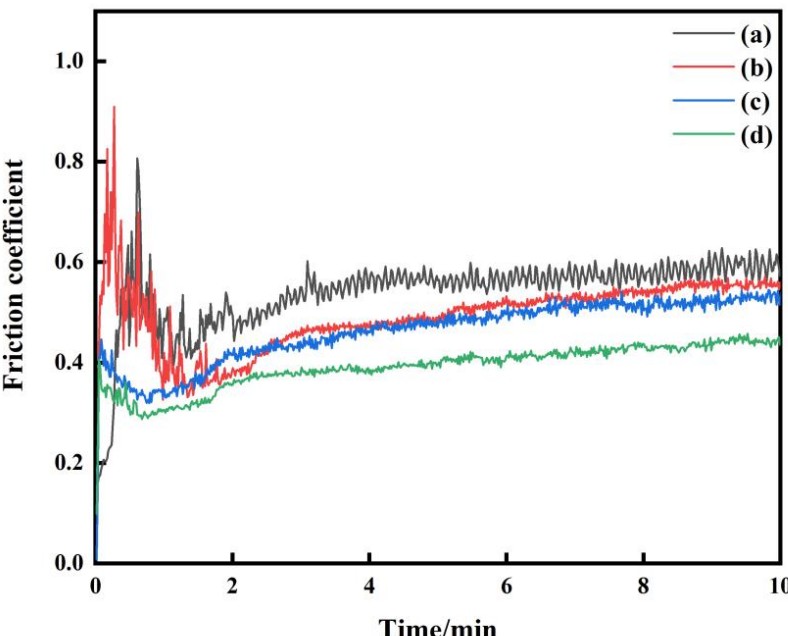

**Figure 8.** Evolution of friction coefficients of composite coatings with the friction time: (**a**) MM powder; (**b**) BM powder milled at 150 rpm; (**c**) BM powder milled at 200 rpm; (**d**) BM powder milled at 250 rpm.

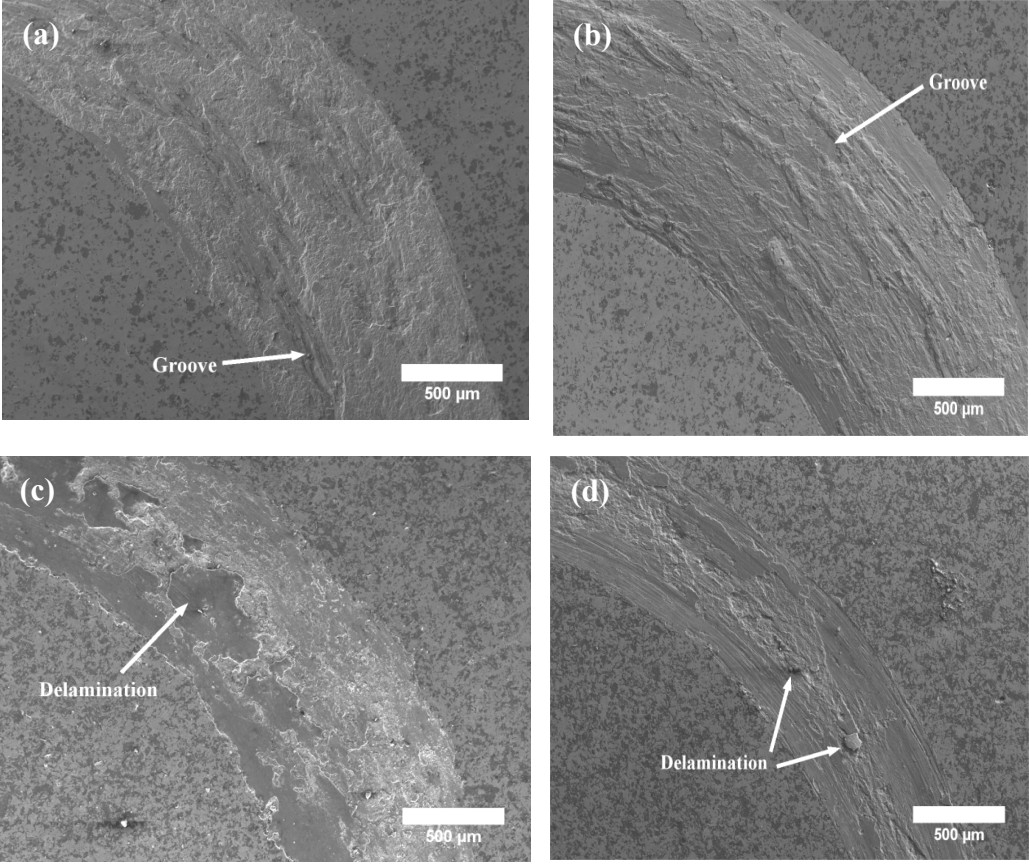

**Figure 9.** SEM images of wear tracks: (**a**) MM powder, (**b**) BM powder milled at 150 rpm, (**c**) BM powder milled at 200 rpm and (**d**) BM powder milled at 250 rpm.

## 4. Conclusions

Copper-based mechanically mixed (MM) and ball-milled (BM) composite Cu-Ni-Al-$Al_2O_3$ coatings were deposited on AZ91D Mg alloy substrates using LPCS. According to the SEM images of BM powders, a core–shell structure with uniformly mixed components formed at the rotation speed above 200 rpm. The relevant coatings exhibited a uniform and dense structure with a low degree of porosity (0.29%) as well as strong mechanical bonding to the substrate. Aside from this, the deposition efficiency of BM powder at 200 rpm could reach 40.61% and exceeded 10.89% than that of MM powder, which was attributed to the ability of the Al shell of the sprayed powder to undergo more severe plastic deformation and deposit into coatings, and the change in microhardness with the coating thickness was the smallest, indicating that the uniform coating composition and the coating hardness was stable at around 153.03 $HV_{0.1}$. The presence of the Al shell in BM powders exerted a positive effect on the adhesive properties of the composite coatings, resulting in the higher adhesion of BM coatings than of MM coatings. Meanwhile, the friction coefficients and WR values of the BM coatings were inferior to those of the MM coatings at rotation speeds of higher than 200 rpm. Based on the structural analysis and mechanical parameters of coatings, the ball milling of spraying powders at 200 rpm ensured the uniform deposition of coatings and enhanced their mechanical characteristics.

**Author Contributions:** Conceptualization, H.L. and S.P.; methodology, M.F.; software, H.Z.; validation, H.L. and M.D.; formal analysis, H.Z.; investigation, H.L.; resources, S.P.; data curation, H.L., C.Z., L.M., X.L. and M.D.; writing—original draft preparation, H.L. and M.D.; writing—review and editing, H.L.; visualization, H.L.; supervision, M.D. and Y.F.; project administration, H.L.; funding acquisition, M.D. All authors have read and agreed to the published version of the manuscript.

**Funding:** This work was funded by the Natural Science Foundation of Heilongjiang Province (LH2021E029).

**Institutional Review Board Statement:** Not applicable.

**Informed Consent Statement:** Not applicable.

**Data Availability Statement:** The data presented in this study are available on request from the corresponding author.

**Conflicts of Interest:** The authors declare no conflict of interest.

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
