# Peer review of "Effect of Ball-Milled Feedstock Powder on Microstructure and Mechanical Properties of Cu-Ni-Al-Al2O3 Composite Coatings by Cold Spraying"

_coatings, doi:10.3390/coatings13050948_

Round 1

Reviewer 1 Report

The Authors investigated the Effect of Ball-Milled Feedstock Powder on Microstructure and Mechanical Properties of Cu-Ni-Al-Al2O3 Composite Coatings by Cold Spraying. However, some of the following points need to be addressed to improve the quality of the paper.

1. Add the novelty statement

2. Include the porosity measurement details in the methodology section

3. In Table-2, The  wear rate (2.47 (×10-12m3/m))  is very low when compared to other specimens. However, same sample shows low hardness (136.55HV). Please check the wear rate and hardness. In general, low hardness exhibits more wear rate

4.  Adhesion of the BM coating was 40.29 MPa at the speed of 200 rpm (Table 2), being 29.26% higher than that of MM coating. whats the reason behind it. Please justify with some references.

5. In Fig. 9, Please indicate the surface impurities like, continuous peeling cracks and delamination with arrow marks in the SEM images.

6. Conclusions should be specific findings of the paper. please edit the conclusions

Reviewer 2 Report

Dear Authors,

I have read your paper "Effect of Ball-Milled Feedstock Powder on Microstructure and Mechanical Properties of Cu-Ni-Al-Al2O3 Composite Coatings by Cold Spraying" carefully.

Explanations are clear and the review is easy to read. 

However, it requires few corrections.

  1. Please add, diameter of the ZrO balls..
  2. Please add, the informations about roughness of the coatings.
  3. Please add, improve figure 9
  4. Please add, general quantitative results in the abstract and conclusions.  .  

The paper can be accepted for publication only after minor improvements.

Reviewer 3 Report

This article aims to study the effect of ball-milled powder morphology on the structural and mechanical characteristics of the cold-sprayed Cu-Ni-Al-Al2O3 composite coatings.

The manuscript is well-written and organized, although some aspects need more attention, and a part of the results should be carefully revised. The current form of the manuscript requires major revisions before publication in the Coatings journal.

Some questions and observations are summarized below:

1.      Please carefully proofread spell check to eliminate grammatical and spelling errors like: line 19: the microstructure, composition, and mechanical properties of the coatings were studied; line 21: the most uniform composition; Line 53: a better wear resistance; line 70: served as substrate; line 81: the phase compositions of spraying powders; line 244: 2.47×10-12m3/m;

2.      The references should be placed in the text before punctuation: line 32: manufacturing. [5-7]; line 35: coating. [8-10]; line 38: coating. [11-13], and so on.

3.      Line 105: ΔV is the volume wear number of composite coatings. Please revise.

4.      Figure 1 and Figure 3: It is not clear why in Figure 1d, the Al corresponds to the brighter areas, while in Figure 3, the Al was indicated in the darker regions of the image. Which type of SEM signal was used to acquire the presented images?

5.      The roughness of the substrate before deposition is not mentioned in the manuscript. Does this characteristic have an influence on the bonding between coating and substrate?

6.      Table 1 summarizes the EDS analysis of the coatings. How did the authors find the weight percent of Al2O3 from this type of analysis? Moreover, the XRD diffraction patterns do not indicate any amount of Al2O3 in the coatings, and the SEM micrographs do not show any Al2O3 particles in the coatings. The authors should revise this aspect.

7.      Table 2: How was the deposition efficiency calculated? Although the size of the particles is higher in the case of 250 rpm powder, the deposition efficiency is lower. Please explain.

8.      How were the samples bonded together for the tensile strength test? Some SEM micrographs for the fracture surfaces would be useful to understand the fracture mechanism.

 The manuscript presents grammatical and spelling error that need to be revised before publication, like: line 19: the microstructure, composition, and mechanical properties of the coatings was studied; line 21: the more uniform composition; Line 53: the better wear resistance; line 70: served as the substrates; line 81: the phase compositions of sprayed powders; line 244: 2.47×10-12m3/m;

Round 2

Reviewer 3 Report

The current form of the manuscript was significantly improved.

There are still some minor spelling issues that need to be addressed, like:

- Line 21: exhibited the most (not the more) uniform composition

- Line 97-98: The sentence needs to be revised. For a better understanding. the authors should use the past tense form of the verbs, exactly like in the description of the other analysis methods.

Minor editing of English is still required.

Author Response

Response to Reviewer Notes

In general, we have paid our close attention to each point raised by the referees. Revision has been carefully made to the manuscript. (Every revision part was marked by red font color in the revised manuscript carefully).

A detailed technical response to reviewers is as follows, with the reviewer’s comment marked by the black font color and the reply marked by blue font color.

  1. Line 21: exhibited the most (not the more) uniform composition

[Reply]: Thank you for your suggestion. In this paper, the relevant sentence was revised to: exhibited the most uniform composition. The corresponding descriptions were provided on page 1.

  1. Line 97-98: The sentence needs to be revised. For a better understanding. the authors should use the past tense form of the verbs, exactly like in the description of the other analysis methods.

[Reply]: Thank you for your suggestion. In this paper, the relevant sentence was revised to: 20g of spray powder was weighed and sprayed onto the AZ91D magnesium alloy substrate, and the powder deposition efficiency was calculated using the following formula. The corresponding descriptions were provided on page 3.